# Prone Positioning Is Safe and May Reduce the Rate of Intubation in Selected COVID-19 Patients Receiving High-Flow Nasal Oxygen Therapy

**DOI:** 10.3390/jcm10153404

**Published:** 2021-07-30

**Authors:** Andrea Vianello, Martina Turrin, Gabriella Guarnieri, Beatrice Molena, Giovanna Arcaro, Cristian Turato, Fausto Braccioni, Leonardo Bertagna De Marchi, Federico Lionello, Pavle Subotic, Stefano Masiero, Chiara Giraudo, Paolo Navalesi

**Affiliations:** 1Department of Cardiac, Thoracic and Vascular Sciences, University of Padova, 35128 Padova, Italy; martina.turrin@aopd.veneto.it (M.T.); gabriella.guarnieri@unipd.it (G.G.); beatrice.molena@unipd.it (B.M.); giovanna.arcaro@aopd.veneto.it (G.A.); fausto.braccioni@aopd.veneto.it (F.B.); leonardo.bertagnademarchi@studenti.unipd.it (L.B.D.M.); federico.lionello@aopd.veneto.it (F.L.); pavle.subotic@studenti.unipd.it (P.S.); 2Department of Molecular Medicine, University of Pavia, 27100 Pavia, Italy; cristian.turato@unipv.it; 3Department of Neurosciences, University of Padova, 35128 Padova, Italy; stefano.masiero@unipd.it; 4Department of Medicine DIMED, University of Padova, 35128 Padova, Italy; chiara.giraudo@unipd.it (C.G.); paolo.navalesi@unipd.it (P.N.)

**Keywords:** COVID-19, prone positioning, acute respiratory failure, endotracheal intubation

## Abstract

Background: Patients with COVID-19 may experience hypoxemic Acute Respiratory Failure (hARF) requiring O_2_-therapy by High-Flow Nasal Cannula (HFNO). Although Prone Positioning (PP) may improve oxygenation in COVID-19 non-intubated patients, the results on its clinical efficacy are controversial. The present study aims to prospectively investigate whether PP may reduce the need for endotracheal intubation (ETI) in patients with COVID-19 receiving HFNO. Methods: All consecutive unselected adult patients with bilateral lung opacities on chest X-ray receiving HFNO after admission to a SARS-CoV-2 Respiratory Intermediate Care Unit (RICU) were considered eligible. Patients who successfully passed an initial PP trial (success group) underwent PP for periods ≥ 2 h twice a day, while receiving HFNO. The study’s primary endpoint was the intubation rate during the stay in the RICU. Results: Ninety-three patients were included in the study. PP was feasible and safe in 50 (54%) patients. Sixteen (17.2%) patients received ETI and 27 (29%) escalated respiratory support, resulting in a mortality rate of 9/93 (9.7%). The length of hospital stay was 18 (6–75) days. In 41/50 (80%) of subjects who passed the trial and underwent PP, its use was associated with clinical benefit and survival without escalation of therapy. Conclusions: PP is feasible and safe in over 50% of COVID-19 patients receiving HFNO for hARF. Randomized trials are required to confirm that PP has the potential to reduce intubation rate.

## 1. Introduction

The SARS-CoV-2 infection, which causes COVID-19, was declared a global pandemic by the World Health Organization (WHO) on 11 March 2020. Patients with COVID-19 may experience severe hypoxemic Acute Respiratory Failure (hARF) consequent to Acute Respiratory Distress Syndrome (ARDS) requiring supportive respiratory therapy [1]. Although presenting high failure rates, the use of O_2_-therapy by High-Flow Nasal Cannula (HFNO) has been suggested to avoid the need for endotracheal intubation (ETI) and invasive mechanical ventilation (IMV) in COVID-19 patients with hARF refractory to conventional oxygen therapy [2,3].

Prone positioning (PP) may affect the physiology of pulmonary gas exchange and improve oxygenation in both intubated and non-intubated patients with ARDS: indeed, making the distribution of ventilation more homogeneous, PP may ameliorate ventilation-perfusion matching and reduce intrapulmonary shunting [4]. Findings from recent studies showed that the adjunctive use of PP may reduce the need for ETI and improve survival in COVID-19 patients with hARF requiring HFNO, with the caveat that a remarkable proportion of subjects may not tolerate PP due to discomfort, anxiety, and the inability to change position [5,6,7,8]. However, the published studies are quite limited in number and confined to small case series, often including patients with different severities of hypoxemia. The present study aims to prospectively investigate whether PP may be feasible and safe, and reduce the need for ETI in spontaneously breathing patients receiving HFNO for severe hARF secondary to COVID-19. In addition, we assessed the rate of escalation of respiratory support, i.e., Noninvasive Ventilation (NIV) or ETI, in-hospital mortality rate, and length of hospital stay.

## 2. Methods

This is a single center, prospective cohort study conducted at the SARS-CoV-2 Respiratory Intermediate Care Unit (RICU) of the University Hospital of Padua between 1 November 2020 and 28 February 2021. All the study’s participants signed general consent for use of de-identified clinical data for research, analysis, and reporting; the data were anonymized by assigning a deidentified patient code. Ethical approval was waived by the local Ethics Committee in view of the fact that the study involved procedures that are part of an internal hospital protocol approved by the Regional Health Authority. The study was carried out in accordance with the Declaration of Helsinki of 1975.

### 2.1. Patients

All consecutive unselected adult patients with bilateral lung opacities on chest X-ray receiving HFNO after admission to the SARS-CoV-2 RICU during the study period were considered eligible. The criteria for patients’ admission to our RICU are: (1) laboratory confirmed COVID-19 infection; and (2) need for noninvasive respiratory support, i.e., HFNO, Continuous Positive Airway Pressure (CPAP) or NIV, to achieve oxygen saturation (SpO_2_) ≥ 92%. Exclusion criteria were: (1) impending ETI and/or hemodynamic instability; (2) severe obesity, as defined by Body Mass Index (BMI) ≥ 30; (3) pregnancy; (4) inability to turn over in bed without assistance; (5) prior treatment with IMV, NIV or CPAP, and (6) patients with Do-Not-Intubate (DNI) order as collegially discussed between pulmonologists and Intensive Care Unit (ICU) physicians at the time of HFNO initiation.

The following patients’ demographic and clinical features were collected at study entry: age, gender, smoking habits, and BMI. At RICU admission the following parameters were recorded for further analysis: days from onset of symptoms; Barthel Index for Activities of Daily Living [9]; Heart Rate (HR); Respiratory Rate (RR); body temperature; leukocyte count; D-dimer; serum C-reactive protein (CRP); arterial PaO_2_; PaCO_2_, and pH during spontaneous breathing with supplemental oxygen; and arterial oxygen tension (PaO_2_) to inspired oxygen fraction (FiO_2_) ratio (PaO_2_/FiO_2_). The ROX index, defined as the ratio of SpO_2_/FiO_2_ to RR, was calculated [10]. Sequential Organ Failure Assessment (SOFA) score [11] and age-adjusted Charlson Comorbidity Index (ACCI) score were also determined for each patient [12]. Chest X-ray was scored based on the extension of ground-glass opacity and consolidations using a previously validated composite COVID-19 chest radiography score (CARE) [13].

### 2.2. Interventions

All patients underwent a trial of PP after being taught and encouraged by RICU staff to lie in a prone position for at least 2 consecutive hours while receiving HFNO. Flow rate and FiO_2_ were not changed during the trial. Over the course of PP trial, ECG, pulse oximetry, invasive and/or non-invasive blood pressure and RR were continuously monitored. SpO_2_ was recorded at baseline (supine position) and 15 min after the patient had been turned to prone position, and the SpO_2_/FiO_2_ (S/F) ratio was calculated. Patients were divided in two groups depending on the result of PP trial, PP success, including patients who successfully passed the trial, and PP failure, including patients who failed the trial. Reasons for failure were considered the following: (1) refusal to consent to testing; (2) inadequate cooperation and/or altered mental status; (3) intolerance to PP, as defined by the incapacity to maintain PP for at least two consecutive hours; and (4) lack of improvement or deterioration in oxygenation.

Patients who successfully passed the PP trial underwent PP for periods ≥ 2 h twice a day, while receiving HFNO. We chose the 2-h minimum proning time in line with a previous study by Halifax et al. [8]. HFNO was also administered outside the PP period. PP sessions were initiated on the first day of RICU admission. HFNO settings were kept constant during PP. PP sessions were regularly performed during RICU stay until the patient was able to maintain SpO_2_ level ≥ 92% by conventional O_2_-therapy. 

Patients who did not pass the PP test underwent HFNO in semirecumbent position. 

HFNO was delivered using the AIRVO_2_ respiratory humidifier (Fisher & Paykel Healthcare, Auckland, New Zealand), with an integrated flow generator able to adjust F_I_O_2_ (between 0.21 and 1.0) and to deliver an air/oxygen mixture at flow rates of up to 60 L/min. The gas mixture is routed through a circuit via large-bore bi-nasal prongs. HFNO was set at 60 L/min gas flow rate, at a temperature of 37°. F_I_O_2_ was adjusted to provide the minimum F_I_O_2_ necessary to maintain a SpO_2_ ≥ 92%. To reduce the risk of viral transmission, the patient wore a surgical mask over the mouth and the nasal prongs. 

All patients received standardized pharmacologic treatment. Corticosteroid therapy was administered according to a protocol (dexamethasone, 6 mg/die up to 10 days); patients also received deep venous thrombosis (DVT) prophylaxis (enoxaparin sodium, 100 IU/Kg/die). When laboratory findings were consistent with a “cytokine storm”, patients were prescribed off-label treatment with Tocilizumab, a recombinant humanized monoclonal antibody directed against the interleukin-6 (IL-6) receptor. Patients admitted to the RICU within 10 days from symptom onset also received the antiviral drug Remdesivir, (200 mg loading dose on day 1, followed by 100 mg daily for up to 9 additional days). Tocilizumab treatment protocol included one dose of 8 mg/kg up to a maximum of 800 mg per dose; a second administration (same dose) was given after 12 h if respiratory function had not recovered. A restrictive fluid management strategy was adopted in all cases to mitigate pulmonary edema.

ECG, pulse oximetry, invasive and/or non-invasive blood pressure and RR were continuously monitored during the RICU stay. Adverse events during PP, including intravenous catheter dislodgement, oxygen cannula removal, vomiting, hemodynamic instability, thromboembolism, or pressure ulcers were recorded.

When SpO_2_ ≥ 92% could not be achieved with FiO_2_ ≤ 0.6, treatment was escalated from HFNO to NIV. The decision to proceed to ETI was agreed between the attending physicians and the Intensive Care team, according to the hospital internal protocol [14].

### 2.3. Outcome Measures and Statistical Analysis

The primary outcome was the rate of ETI secondary endpoints were the following: (1) rate of escalation of respiratory support, i.e., NIV or ETI; (2) in-hospital mortality rate; (3) length of hospital stay. The outcomes were censored on 31 March 2021 for patients still hospitalized on that day. The normality of data distribution was tested using the Kolmogorov–Smirnov test. The independent unpaired Student’s *t* test was used to compare normally distributed continuous variables; nonparametric data were compared using the Wilcoxon rank-sum test. Categorical variables were compared using the Chi-squared test or Fisher’s Exact Test, as indicated. Survival from the time of admission to RICU was calculated using the Kaplan–Meier method; the log rank test was used to compare survival curves between groups. Two-tailed *p* values < 0.05 were considered significant. All statistical analyses were conducted using GraphPad Prism version 8.02 (GraphPad Software, San Diego, CA, USA) or MedCalc® Statistical Software version 20 (MedCalc Software Ltd, Ostend, Belgium).

## 3. Results

### 3.1. Feasibility

As shown in Figure 1, throughout the study period (from 1 November 2020 to 28 February 2021) 310 patients were admitted to RICU for severe hARF secondary to COVID-19 not responding to conventional O_2_-therapy. One hundred and five (33.9%) had received prior treatment with NIV and/or IMV in the ICU setting; 22 (7.1%) received NIV at admission, and the other 183 (59%), were prescribed HFNO as the first-line respiratory support; 93 out of 183 were prescribed HFNO with no ceiling of treatment and were eligible for enrollment in our study. Fifty patients (54%) successfully passed PP trial and were included in the PP success group. In this group, S/F ratio significantly improved between supine and prone positioning (158 (91–271) vs. 164 (96–280); *p* < 0.0001). The remaining 43 patients (46%) failed the trial because of consent refusal (12 patients), inadequate cooperation (14 patients), and discomfort during positioning (17 patients) consequent to back or shoulder pain (15 patients) and dizziness (2 patients). As shown in Table 1, baseline characteristics, clinical and laboratory data at RICU admission were not significantly different between the two groups, except for the CARE score that was slightly, though significantly, worse in the PP success group (18 (12–86) vs. 16 (10–22); *p* < 0.0001). The number of patients receiving treatment with Tocilizumab and/or Remdesivir did not differ between groups (3/50 vs. 1/43; *p* = 0.6211; and 26/50 vs. 16/43; *p* = 0.2100, for successes and failures, respectively). Among patients who successfully passed PP trial, the median number of PP sessions was 6 (2–27), with 41 subjects undergoing PP for 2 to 10 sessions, 7 for 11 to 20 sessions and 2 for over 20 sessions, while the median duration for each cycle was 2 (2–6) hours, with a maximum duration of 2, 4 and 6 h in 42, 6 and 2 patients, respectively.

### 3.2. Safety

No adverse event was recorded during application of PP.

### 3.3. Efficacy

As presented in Table 2, 16 (17.2%) patients received ETI and 27 (29%) escalated respiratory support, resulting in a mortality rate of 9/93 (9.7%). The length of hospital stay was 18 (6–75) days. In selected patients who regularly performed PP, intubation rate was 8% and its use was associated with clinical benefit and survival without escalation of therapy in over 80% of cases. A relationship between the amount of PP and its effect on patients’ outcomes was not demonstrated.

The median duration of the follow-up period was 101 (11–156) days. The log-rank test showed that patients in the PP success group survived for a significantly longer time (150.98 ± 3.49 days) compared to those in the PP failure group (138.61 ± 4.5 days; (*p* = 0.0026)) (Figure 2). At the end of the follow-up period, the rate of survival was 96.0% ± 2.77% in the PP success and 73.5% ± 6.89% in the PP failure group. Hazard Ratio (HR) for death was 0.1834 (95% CI, 0.06086 to 0.5529).

## 4. Discussion

In the present study, PP was feasible and safe in slightly more than 50% of patients receiving HFNO. In selected subjects who regularly performed PP, this treatment was associated with clinical benefit and survival without escalation of therapy, showing the potential for improving the outcome of patients with hARF secondary to COVID-19.

A number of physiologic studies showed improvement of physiological endpoints associated with PP, both in intubated [4] and spontaneously breathing patients with COVID-19 [15]. The improvement of physiological endpoints, however, may not translate into better clinical outcomes. Previous results on the clinical efficacy of PP in COVID-19 patients receiving HFNO are controversial: whilst small case series showed promising results, suggesting that successful PP can reduce the risk of ETI [5,6,7,8], a recent multicenter, prospective cohort study concluded that awake PP does not significantly reduce the risk of ETI [16].

These conflicting results may depend on a variety of reasons. First of all, the lack of standardized criteria for ETI, which was overcome in the present investigation by the use of a standardized protocol, based on stepwise utilization of HFNC, NIV and ETI [14,17]. Differences in timing of PP initiation and uneven duration and frequency of PP sessions may also contribute to explain these discrepancies. In the present study, we applied PP in the first day of RICU admission: early PP initiation might have played an important role, as COVID-19 patients are more likely to respond to pronation in the early phases [18,19]. Compared to late application, early PP is better tolerated, results in better oxygenation, and prevents further disease progression [20].

Patients’ older age and more severe clinical conditions could also contribute to explain these differences.

In support of the potential benefit from PP, patients in the PP success group survived for a significantly longer time compared to those in the PP failure group (Figure 2): however, this result is likely to be flawed by non-random allocation to “success” and “failure” arms, which was performed according to physicians’ assessment of patient’s response to the initial PP trial.

With regard to proning time, Xu et al. proposed a prolonged time for PP sessions also in awake patients (>16 h/day) in order to obtain a significant survival benefit [6]. In our study, we set a minimum duration of two hours for the PP sessions with no maximum limit and obtained a median PP session duration of 2 (2–6) hours, which was, nonetheless, sufficient to achieve clinical benefits. This result is in line with data from Halifax et al. showing the positive effect of 2-h sessions on in-hospital mortality rate of patients with COVID-19 requiring respiratory support [8].

Our study has two major limitations. First, the study was conducted in a single center, raising doubts on the possibility of generalizing our results. Indeed, we share this limit with most previous studies evaluating PP in spontaneously breathing patients, with or without noninvasive respiratory support. The second important limitation is unavailability of some relevant data, such as sodium and fluid intake and fluid balance. 

## 5. Conclusions

PP was feasible and well tolerated without short-term side effects in 54% of patients receiving HFNO for hARF secondary to COVID-19. In selected patients, its use was associated with clinical improvement and survival without escalation of therapy.

Future randomized trials are required to confirm that PP has the potential to improve the outcomes of patients with hARF secondary to COVID-19 and ascertain whether these results may be attainable in hARF of different etiologies.

## Figures and Tables

**Figure 1 jcm-10-03404-f001:**
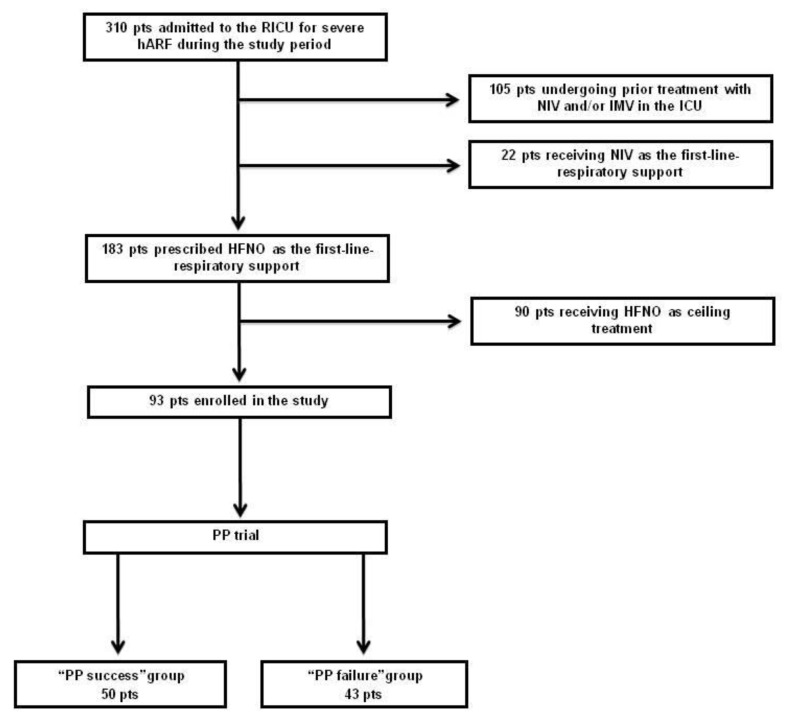
Study profile (hARF = hypoxemic Acute Respiratory Failure; HFNO= High Flow Nasal Oxygen; IMV= Invasive Mechanical Ventilation; NIV= Noninvasive Ventilation; PP = Prone Positioning; RICU = Respiratory Intermediate Care Unit).

**Figure 2 jcm-10-03404-f002:**
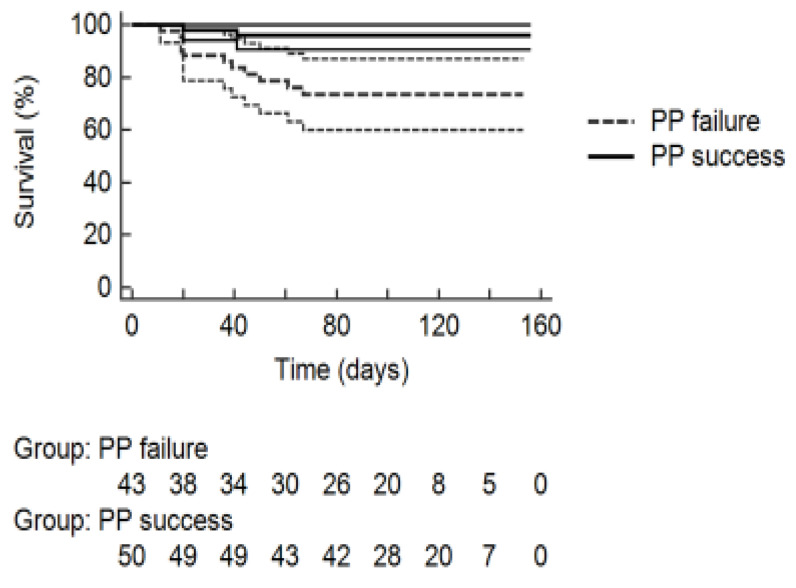
Kaplan–Maier estimates of survival function after Respiratory Intermediate Care Unit admission, stratified according to the group of origin. (PP = Prone Positioning).

**Table 1 jcm-10-03404-t001:** Patients’ baseline demographic and clinical characteristics, and clinical and laboratory data at RICU admission. *p*-values refer to differences between Prone Positioning (PP) success and failure groups. (ACCI = Age-adjusted Charlson Comorbidity Index; BMI = Body Mass Index; CRP = C-Reactive Protein; PaO_2_/FiO_2_ = arterial oxygen tension to inspired oxygen fraction ratio; SaO_2_ = arterial oxygen saturation; SOFA = Sequential Organ Failure Assessment; WBC = White Blood Cell).

	All Cases (*n* = 93)	PP Success (*n* = 50)	PP Failure (*n* = 43)	*p*-Value
**Baseline demographic and clinical data**				
Age (yrs), median (range)	68(36–89)	67 (36–89)	69 (37–86)	0.37
Gender (M/F)	59/34	33/17	26/17	0.66
Smokers, N	17	9	8	0.99
BMI (kg/m^2^), median (range)	28.0 (20.8–41.5)	26.9 (20.8–41.5)	28.3 (22.9–33.3)	0.75
**Clinical, laboratory and blood gas data at RICU admission**				
Time since symptom onset (days), median (range)	4 (3–16)	4 (3–10)	4 (3–16)	0.92
Barthel index, median (range)	50 (4–100)	45 (10–100)	60 (4–100)	0.08
Heart rate (beats/min), median (range)	80 (55–126)	80 (55–110)	80 (60–126)	0.29
Respiratory rate (breaths/min), median (range)	22 (13–39)	22 (13–39)	22 (15–32)	0.69
Pts with fever, N. (Temperature > 38 °C)	13	8	5	0.76
Total WBC count (×10^9^/L), median (range)	8.6 (1.6–23.9)	8.3 (1.6–23.9)	8.7 (2.4–17.2)	0.88
Pts with D-dimer level above the normal range, n	38	22	16	0.09
Serum CRP (μg/mL), median (range)	96.0 (11.0–270.0)	84.2 (11.0–230.0)	99.5 (13.0–270.0)	0.23
PaO_2_ * (mmHg), median (range)	68.0 (6.8–157.6)	64.9 (6.8–157.6)	68.8 (39.2–110.9)	0.99
PaCO_2_ (mmHg), median (range)	33.4 (23.6–70.0)	32.7 (23.6–60.0)	34.9 (25.2–70.0)	0.23
Arterial pH, median (range)	7.46 (7.25–7.56)	7.46 (7.25–7.56)	7.47 (7.32–7.53)	0.79
SaO_2_ (%), median (range)	95 (84–100)	94 (89–100)	95 (84–100)	0.93
PaO_2_/FiO_2_, median (range)	101.8 (6.8–300.0)	107.2 (6.8–300.0)	92.4 (52.4–240.9)	0.35
ROX index, median (range)	5.65 (2.41–22.06)	5.81 (2.41–21.50)	4.94 (2.97–22.06)	0.37
SOFA score, median (range)	2 (1–5)	2 (1–5)	2 (2–5)	0.01
Pts with co-morbidities, N	87	45	42	0.21
ACCI	4 (0–13)	4 (0–13)	4 (0–8)	0.82
CARE score, median (range)	18 (10–86)	18 (12–86)	16 (10–22)	<0.0001

* with supplemental oxygen.

**Table 2 jcm-10-03404-t002:** Patients’ outcomes following RICU admission. (ETI= Endotracheal Intubation; PP = Prone Positioning). *p*-values refer to differences between Prone Positioning (PP) success and PP failure groups.

	All Cases (*n* = 93)	PP Success (*n* = 50)	PP Failure (*n* = 43)	*p*-Value
Patients receiving ETI, *n* (%)	16 (17)	4 (8)	12 (28)	0.014
Patients escalating respiratory support, *n* (%)	27 (29)	9 (18)	16 (37)	0.059
Death during hospitalization, *n* (%)	9 (10)	2 (4)	7 (16)	0.047
Length of hospital stay (days), median (range)	18 (6–75)	17 (6–46)	21 (7–75)	0.001

## Data Availability

The data presented in this study are available on request from the corresponding author. The data are not publicly available due to privacy restrictions.

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
