# Peer review of "Prone Positioning Is Safe and May Reduce the Rate of Intubation in Selected COVID-19 Patients Receiving High-Flow Nasal Oxygen Therapy"

_jcm, 2021, doi:10.3390/jcm10153404_

Round 1
Reviewer 1 Report
Review
Title: Prone positioning reduces the rate of intubation in COVID-19 patients receiving high-flow nasal oxygen therapy.
Summary:
The authors are to be congratulated for undertaking a detailed investigation of nearly 100 patients with hypoxic respiratory failure secondary to SARS-Cov19 infection to test the benefit of prone position spontaneous breathing (PPSB). There is much anecdotal and published evidence to support this hypothesis. The authors conclude that prone position spontaneous breathing has the potential to improve the outcome of Covid-19 patients with moderately severe hypoxia if commenced within first 24-hours of ICU admission
The authors screened over 300 patients, and enrolled 93 (32%) in their trial. All subjects consented and underwent PPSB trial and based on their response were dichotomised into PPSB-success subgroup and PSB-failure group. The authors compared the outcome of these two subgroups and provide very detailed demographic and clinical information about all patients.
The authors correctly conclude that PPSB is feasible and safe in 55% of subjects; and requires dedicated staff with PPSB expertise; and in selected patients PP has the potential for improving patient outcome; and that randomized trials are required to confirm these results.
Detailed comments and recommendations.
- Methodology
The authors acknowledge that their investigation was an unblinded observational single centre study and was not a randomised-controlled trial. However, they appear to have analysed and reported patient outcomes as if it was a randomised-controlled trial. They compare the two subgroups as if they were randomly allocated to either subgroup. This is misleading and incorrect.
The two subgroups were highly selected. First the authors screened only those patients admitted to the RICU. From 310 screened subjects they rejected 70% (n=217) and enrolled 93 subjects based on reasonable clinical criteria. Finally the authors selected which subgroup each subject would be allocated to based on the initial response to the asme treatment that was under investigation.
- Subgroup Allocation
Each subject was selectively allocated to one of two subgroups according to the authors assessment of the patient’s response to a trial of PPSB. Most of those allocated to the non-PPSB “control” group were allocated on the basis of non-compliance and not because of failure of the therapy, per se. This allocation process was unblinded and determined by the authors and not by random allocation.
It is not surprising to find that patients who were given treatment (PPSB) did better than those who are not given (the same) treatment (“control" group). The authors withheld PPSB from those that they already knew would not benefit from PPSB. And they asigned to the treatment group on those would were likely to benefit. The authors admit that “this approach made more likely a positive response to treatment”.
- Recommended approach
If the authors intended to compare two groups of patients they should have allocated PPSB by random allocation method and analysed the data on an intention-to-treat classification.
The authors investigation is very similar to a Phase-2 human safety trial of PPSB. Based on their results it would be reasonable to proceed to a phase-3 randomised controlled trials to test efficacy of PPSB. The evidence they provide supports this conclusion.
- Adverse Events
The claim that “No adverse event was recorded during PP sessions.” seems to be at odds with the evidence and misleading. A high proportion of the 93 study subjects required escalation of therapy, intubation and IMV, or died.
I presume the comment refers to the list of side-effects intravenous catheter dislodgement, oxygen cannula removal, vomiting, hemodynamic instability, thromboembolism, pressure ulcers. Is this correct?
- Alternative Conclusion
Based on the evidence presented we may conclude that PPSB was associated with
- Clinical improvement and survival (without escalation of therapy) in 45% of subjects.
- High rate of failure from poor patient tolerance and compliance (28%) and intubation (36%).
- 20% mortality rate, which is similar to outcome of intubated COVID19 subjects.
6. The authors should clearly state all limitations, including:
- Unblinded trial design leading to potential for bias and confounding.
- Non-random (and selective) allocation to trial and control arms, according to author assessment of benefit from PSB.
- High risk of Type1 and 2 statistical errors due to low sample size. See below.
- Allocation of subgroup and escalation of treatment (NIV, IMV) where determined by the authors.
- Missing data: sodium and fluid intake and fluid balance,
7. Conclusions
Based on the selection method and the evidence provided the authors cannot conclude that PPSB “reduced the need for ETI and shorter LOS”. The available evidence does support the following conclusion: PPSB has a high failure rate due to poor compliance (28%) or clinical failure (36% intubation) resulting in escalation of respiratory support in 55%, and a mortality rate of 20%.
8. Ethical concern.
The authors description of patient consent and Hospital Ethics Committee decision is confusing. They state that patients “gave informed consent” but in the next sentence also state that “ethical approval was waived”. Can the authors please clarify?
I presume that patients gave general consent for use of de-identified clinical data for research, analysis, and reporting but the consent process did not specify this PPSB study in particular. And that the Hospital Ethics Committee waived the need for consent to be enrolled in this specific study. Is this correct?
9. Power and sample size.
This observational study appears to be under-powered and thus cannot answer the research hypothesis with much accuracy. This should be noted in the list of limitation.
If we assume (from the reported rates) that annual RICU throughput is 1,000 patients (310 every 4-months) and that the baseline mortality is 10% (9 deaths in 93 subjects), then a sample size of over 200 subjects is required to identify a change in mortality rate (in the PPSB treatment group) with 90% certainty and 5% margin of error. Therefore this study is likely to be under-powered.
10. Statistical Frailty
This study group is likely to have considerable heterogeneity and display statistical frailty. The numbers included in this study are small and subject to Type 1 error. The authors should include this in the list of limitations.
For example, with the addition of only two deaths in the PPSB subgroup or a reduction of 2 deaths in the control group the outcome differences are no longer significant.
Minor points
11. Figure 2
The figure should include a table of the number of cases at each time point beneath the graph. The lines should include 95% confidence interval.
12. Data format.
Fractions to the second or third decimal place is sufficient; four decimal places is unnecessary.
13. Missing Confounders
There are several missing confounder. It would be helpful to know the fluid input and fluid balance of each group.
14. Compliance
It would be helpful to know if there were any preventable causes of poor compliance with PPSB.
15. Dose of PPSB
The authors provide no evidence to justify the daily duration of PPSB and whether or not there is a dose-response curve.
I presume the authors mean Wilcoxon rank-sum test not Wilcoxon exact test in the statistical analysis.
In summary I recommend the manuscript be re-written from the perspective of a phase-2 safety trial.
Author Response
REVIEWER 1
We thank the reviewer for the suggestions provided and the helpful comments.
- Methodology
The authors acknowledge that their investigation was an unblinded observational single centre study and was not a randomised-controlled trial. However, they appear to have analysed and reported patient outcomes as if it was a randomised-controlled trial. They compare the two subgroups as if they were randomly allocated to either subgroup. This is misleading and incorrect.
The reviewer is clearly right when criticizing the non-random allocation of our patients. Worth remarking, however, the “success” and “failure” groups were extremely similar from a clinical and laboratory perspective. Nonetheless, we now add the non-random allocation among the study limitations. Regarding the unblinded study design, frankly speaking, we do not see how we could blind the two groups.
The two subgroups were highly selected. First the authors screened only those patients admitted to the RICU. From 310 screened subjects they rejected 70% (n=217) and enrolled 93 subjects based on reasonable clinical criteria. Finally the authors selected which subgroup each subject would be allocated to based on the initial response to the asme treatment that was under investigation.
We agree with the reviewer that our patients were highly selected; in fact, the aim of our study was just demonstrating the efficacy of PPSB “in a selected population of patients” undergoing HFNO and successfully passing a 2-hr PP trial. For this reason, our exclusion criteria lead to rejection of approximately two-thirds of patients admitted to our RICU. We do not pretend to provide a response on the efficacy of PPSB in the general population of patients with Covid-19. We now make this point clearer by modifying the study title.
- Subgroup Allocation
Each subject was selectively allocated to one of two subgroups according to the authors assessment of the patient’s response to a trial of PPSB. Most of those allocated to the non-PPSB “control” group were allocated on the basis of non-compliance and not because of failure of the therapy, per se. This allocation process was unblinded and determined by the authors and not by random allocation.
It is not surprising to find that patients who were given treatment (PPSB) did better than those who are not given (the same) treatment (“control" group). The authors withheld PPSB from those that they already knew would not benefit from PPSB. And they assigned to the treatment group on those would were likely to benefit. The authors admit that “this approach made more likely a positive response to treatment”.
Never claimed this was a control group. Worth noting, “control” (failure) group did not include patients who would not reasonably benefit from PPSB, but rather those who did not pass the PP trial for different reasons and therefore did not receive PPDB. We compared these patients with selected candidates who received the treatment. Non-compliance means failure in this specific case.
- Recommended approach
If the authors intended to compare two groups of patients they should have allocated PPSB. by random allocation method and analysed the data on an intention-to-treat classification.
The authors investigation is very similar to a Phase-2 human safety trial of PPSB. Based on their results it would be reasonable to proceed to a phase-3 randomised controlled trials to test efficacy of PPSB. The evidence they provide supports this conclusion.
Because of the study design, it is impossible to analyse these data on an intention-to-treat basis. We agree that a RCT would be helpful and we also agree that our data support designing a future RCT.
- Adverse Events
The claim that “No adverse event was recorded during PP sessions.” seems to be at odds with the evidence and misleading. A high proportion of the 93 study subjects required escalation of therapy, intubation and IMV, or died.
I presume the comment refers to the list of side-effects intravenous catheter dislodgement, oxygen cannula removal, vomiting, hemodynamic instability, thromboembolism, pressure ulcers. Is this correct?
Correct. This point has been clarified in the “Results” section.
- Alternative Conclusion
Based on the evidence presented we may conclude that PPSB was associated with
- Clinical improvement and survival (without escalation of therapy) in 45% of subjects.
- High rate of failure from poor patient tolerance and compliance (28%) and intubation (36%).
- 20% mortality rate, which is similar to outcome of intubated COVID19 subjects.
Again, the aim of our study was demonstrating the efficacy of PPSB “in a selected population of patients” undergoing HFNO and successfully passing a 2-hr PP trial, not in the general population of patients hospitalized for severe ARF consequent to Covid-19.
To be exact, mortality rate in the whole population was 9.6% (9/93 patients).
- The authors should clearly state all limitations, including:
- Unblinded trial design leading to potential for bias and confounding.
- Non-random (and selective) allocation to trial and control arms, according to author assessment of benefit from PSB.
- High risk of Type1 and 2 statistical errors due to low sample size. See below.
- Allocation of subgroup and escalation of treatment (NIV, IMV) where determined by the authors.
- Missing data: sodium and fluid intake and fluid balance,
The reviewer is right. We expanded the list of study limitations in the “Discussion” section.
- Conclusions
Based on the selection method and the evidence provided the authors cannot conclude that PPSB “reduced the need for ETI and shorter LOS”. The available evidence does support the following conclusion: PPSB has a high failure rate due to poor compliance (28%) or clinical failure (36% intubation) resulting in escalation of respiratory support in 55%, and a mortality rate of 20%.
In our study, poor compliance was just an exclusion criteria to identify patients who were not “the right candidates” to PPDB (“failure” group).
To be exact, mortality rate in the whole population was 9.6% (9/93 patients).
- Ethical concern.
The authors description of patient consent and Hospital Ethics Committee decision is confusing. They state that patients “gave informed consent” but in the next sentence also state that “ethical approval was waived”. Can the authors please clarify?
I presume that patients gave general consent for use of de-identified clinical data for research, analysis, and reporting but the consent process did not specify this PPSB study in particular. And that the Hospital Ethics Committee waived the need for consent to be enrolled in this specific study. Is this correct?
We agree with the reviewer that this point is not clear and have modified the text accordingly
- Power and sample size.
This observational study appears to be under-powered and thus cannot answer the research hypothesis with much accuracy. This should be noted in the list of limitation.
If we assume (from the reported rates) that annual RICU throughput is 1,000 patients (310 every 4-months) and that the baseline mortality is 10% (9 deaths in 93 subjects), then a sample size of over 200 subjects is required to identify a change in mortality rate (in the PPSB treatment group) with 90% certainty and 5% margin of error. Therefore this study is likely to be under-powered.
We conducted a sample size estimation with a type 1 error of 0.05 and a power of 80% determining that a significant clinical difference in endotracheal intubation rate (the study primary endpoint) would be detected with a minimum of 43 subjects per group based on an expected difference in clinical outcome of 0.2 between the “success” and the “failure” group. We nevertheless included statistical frailty in the list of study limitations.
- Statistical Frailty
This study group is likely to have considerable heterogeneity and display statistical frailty. The numbers included in this study are small and subject to Type 1 error. The authors should include this in the list of limitations.
For example, with the addition of only two deaths in the PPSB subgroup or a reduction of 2 deaths in the control group the outcome differences are no longer significant.
We included statistical frailty in the list of study limitations.
Minor points
- Figure 2
The figure should include a table of the number of cases at each time point beneath the graph. The lines should include 95% confidence interval.
Agreed and modified accordingly
- Data format.
Fractions to the second or third decimal place is sufficient; four decimal places is unnecessary.
Agreed and modified accordingly.
- Missing Confounders
There are several missing confounder. It would be helpful to know the fluid input and fluid balance of each group.
Detailed data on fluid balance were not available. However, a restrictive fluid management strategy was adopted in all cases to mitigate pulmonary edema. This information has been added in the “Methods” section.
- Compliance
It would be helpful to know if there were any preventable causes of poor compliance with PPSB.
Based on pre-existing data, a significant proportion of patients could be expected not to tolerate PP, due to discomfort, anxiety, and the inability to change position. This information has been added in the “Introduction” section.
- Dose of PPSB
The authors provide no evidence to justify the daily duration of PPSB and whether or not there is a dose-response curve.
We chose an approach aiming for at least 2-hr proning in line with previous study (Ref 8 in the revised manuscript), showing the positive effect of 2-hr sessions on in-hospital mortality rate. A short comment on this point has been added in the “methods” section. In the great majority of cases (44/50), the maximum duration of PP session was 2 hrs and we could not find a clear relationship between the amount of PP and its effect on patients’ outcome. This is now clearly specified in the “Results” section.
I presume the authors mean Wilcoxon rank-sum test not Wilcoxon exact test in the statistical analysis.
Correct. We modified the text accordingly in the “Methods” section.
Reviewer 2 Report
Thank you for offering me the chance to review your manuscript on prone positioning and its effect on ETI reduction in COVID-19 patients receiving high-flow nasal oxygen therapy.
The topic is of interest, though extensively analyzed in the last months. Here are some minor points.
Why did you chose a 2 h PP interval? From my point of view, this is short (if you look at other recommendations for PP for 12 - 16 hours, e.g. in the Xu article). If described as >= 2 hours, this could mean anything from 2 h 1 min to 24 hours - please specify.
Result section
105 (33.9%) - do not start sentences with numbers, use "one hundred and five" instead.
"The median number of PP sessions per patient was 6 (2-27), while the median duration for each cycle was 2 (2-6) hours. " - again, very short duration? How was the distribution?
PP intervals of "only" 2 hours seem very short to me - I would recommend working this up a little more. For me it seems fascinating that durations that short also help to circumvent escalation (and sometimes futile) therapy. This is a major finding!
Thank you very much for your work - good luck!
Author Response
We thank the reviewer for the suggestions provided and the helpful comments.
Why did you chose a 2 h PP interval? From my point of view, this is short (if you look at other recommendations for PP for 12 - 16 hours, e.g. in the Xu article). If described as >= 2 hours, this could mean anything from 2 h 1 min to 24 hours - please specify.
We chose the 2-hour minimum proning time in line with previous work (Ref 8 in the revised manuscript), showing the positive effect of 2-hr sessions on in-hospital mortality. In particular, PP sessions achieved a maximum duration of 2, 4 and 6 hours in 42, 6 and 2 cases, respectively. This is now clearly specified in the “Results” section
Result section
105 (33.9%) - do not start sentences with numbers, use "one hundred and five" instead.
Agreed and modified accordingly.
"The median number of PP sessions per patient was 6 (2-27), while the median duration for each cycle was 2 (2-6) hours. " - again, very short duration? How was the distribution?
41 subjects underwent PP for 2 to 10 sessions, 7 for 11 to 20 sessions and 2 for over 20 sessions, while the maximum duration for each cycle was 2, 4 and 6 hours in 42, 6 and 2 patients, respectively. These data are now clearly reported in the “Results” section.
PP intervals of "only" 2 hours seem very short to me - I would recommend working this up a little more. For me it seems fascinating that durations that short also help to circumvent escalation (and sometimes futile) therapy. This is a major finding!
More detailed data on the distribution and duration of PP sessions have been reported in the “Results” section. A short comment has been added in the “Discussion” section.
Round 2
Reviewer 1 Report
Thank you for the opportunity to review the manuscript revision.
The authors have made minor changes to the manuscript but have not addressed the major flaw in their methodology and analysis and interpretation. A number of the major concerns raised in the original manuscript have not been adequately addressed.
Based on the results provided the authors can only conclude that :
1. PP is feasible and safe in 55% of subjects and requires dedicated staff with PP expertise.
2. In highly selected patients PP may improve patient outcomes
3. These results do not prove that PP is beneficial.
3. Multicentre randomized trials are required to confirm these results.
4. These results do not prove that PP reduces the need for ETI.
The authors have concluded #1 and #5 but not the others.
The evidence provided does NOT support the conclusions that
1. PP significantly decreased the need for intubation.
2. PP patients “required fewer days of hospital stay”.
3. “in selected patients PP has the potentials[sic] for improving the outcome of patients with hARF secondary to Covid-19”
These conclusions cannot be supported by the research methods they used.
The authors selected the study patients by excluding patients they knew would not benefit from PP. They manipulated the outcome by manipulating (selecting) which groups patients were allocated to.
For example, a more accurate title would be:
Prone positioning appears safe and may reduce the rate of intubation in selected COVID-19 patients.
For example, the authors have correctly noted that their study is limited by non-random allocation to “success” and “failure” groups. This statement is true but they have not changed the statistical analysis to demonstrate they understand this. As stated in the previous review, the authors have analysed the data using statistics that assume patients were randomly allocated to PP treatment or non-PP (control) groups.
If the authors should have randomised patients to PP treatment or non-PP (control) groups before the 2-hour trial of PP. But they did not do this. They selected the patients for each group after a 2-hour trial of PP. This is not random allocation; this is selective allocation. Therefore the authors cannot state that PP “significantly” improves or changes any outcome. The authors can only conclude that PP may benefit patients.
I hope these comments are helpful?
This was the major concern included in the first review. It has not been addressed in the (minor) changes added to the revised manuscript. The authors describe good medicine but very poor statistical analysis and false conclusions. I am not convinced that the authors understand the error in their logic.
Author Response
We thank the reviewer for the suggestions provided and the helpful comments.
We have made every effort to take all of the reviewer’s suggestions concerning methodology, data analysis and interpretation into consideration and modified extensively the study design and conclusions.
We hope that the revised version that we are now submitting will satisfy the reviewer’s concerns.